# Laboratory validation of a clinical metagenomic next-generation sequencing assay for respiratory virus detection and discovery

Tools for rapid identification of novel and/or emerging viruses are urgently needed for clinical diagnosis of unexplained infections and pandemic preparedness. Here we developed and clinically validated a largely automated metagenomic next-generation sequencing (mNGS) assay for agnostic detection of respiratory viral pathogens from upper respiratory swab and bronchoalveolar lavage samples in <24 h. The mNGS assay achieved mean limits of detection of 543 copies/mL, viral load quantification with 100% linearity, and 93.6% sensitivity, 93.8% specificity, and 93.7% accuracy compared to gold-standard clinical multiplex RT-PCR testing. Performance increased to 97.9% overall predictive agreement after discrepancy testing and clinical adjudication, which was superior to that of RT-PCR (95.0% agreement). To enable discovery of novel, sequence-divergent human viruses with pandemic potential, de novo assembly and translated nucleotide algorithms were incorporated into the automated SURPI+ computational pipeline used by the mNGS assay for pathogen detection. Using in silico analysis, we showed that after removal of all human viral sequences from the reference database, 70 (100%) of 70 representative human viral pathogens could still be identified based on homology to related animal or plant viruses. Our assay, which was granted breakthrough device designation from the US Food and Drug Administration (FDA) in August of 2023, demonstrates the feasibility of routine mNGS testing in clinical and public health laboratories, thus facilitating a robust and rapid response to the next viral pandemic.

Respiratory infections are among the most common infections globally and are associated with significant morbidity and mortality[1–3]. Despite their importance, half of adult patients hospitalized in the United States with community-acquired pneumonia, which is most commonly caused by respiratory viruses, have no causative pathogen identified[2–5]. Respiratory infections caused by viruses can be especially challenging to diagnose because of the diversity of potential agents[6–8]. In particular, emerging pandemic viruses represent an unpredictable

threat which traditional diagnostic tools such as nucleic acid amplification tests have not been designed to detect[9]. The importance of unbiased assays for rapid identification of viral pathogens, especially those with sequence-divergent genomes, became evident during the discovery of SARS-CoV-2[10,11].

Metagenomic next-generation sequencing (mNGS) has emerged as an attractive diagnostic method for identifying causative agents in unexplained infections as it provides a comprehensive and agnostic

✉e-mail: charles.chiu@ucsf.edu

approach by which all potential pathogens can be identified in a single assay without the need for specific primers and probes[12,13]. mNGS has been used for broadly diagnosing infections, whether viral, bacterial, fungal, or parasitic, from multiple specimen types[14–16], and its clinical utility has been demonstrated for neurological and bloodstream infections[16–18]. However, despite the favorable performance of mNGS testing as shown by multiple studies, general adoption of mNGS technologies in clinical microbiology laboratories has been hindered by high costs, complex protocols, lack of automation, insufficient standardization of bioinformatic pipelines, prolonged turnaround times (24–72 h), lack of regulatory guidelines for clinical validation, and overall lower sensitivity for detection of common pathogens relative to targeted approaches such as polymerase chain reaction (PCR) assays[19].

Here we describe the development, optimization, and clinical validation of a streamlined and largely automated mNGS laboratory-developed test (LDT) with a sample-to-result turnaround time of less than 24 h for identification of common as well as unexpected and/or novel viral respiratory pathogens. The computational SURPI+ pipeline used by the mNGS assay was modified to provide enhanced analysis capabilities, including viral load quantification, incorporation of curated reference genome databases such as FDA dAtabase for Reference Grade micrObial Sequences (FDA-ARGOS), and sensitive identification of novel, sequence-divergent viruses by de novo assembly and translated nucleotide alignment. We comprehensively evaluated assay performance metrics, including limits of detection, linearity, precision, inclusivity and exclusivity, contamination, interference, matrix effect, stability, accuracy, and capacity to detect novel viruses.

## Results

### Development and optimization of an mNGS assay for detection of viral respiratory pathogens

We developed an mNGS assay for the detection of viral pathogens from respiratory secretions, including upper respiratory swab and bronchoalveolar lavage (BAL) fluid samples (Fig. 1). We leveraged our 7-year experience running clinical mNGS assays for pathogen detection from cerebrospinal fluid[20] by optimizing the sample preparation and bioinformatics analysis protocols to maximize sensitivity and decrease assay sample-to-result turnaround time. We tested different combinations of centrifugation, heat, and addition of a DNA/RNA stabilization medium prior to total nucleic acid extraction and found that centrifugation alone produced the highest yield of detected viral reads. To decrease turnaround times, we used a 15-min protocol for human rRNA depletion and reduced incubation times for the reverse transcription and second-strand cDNA synthesis steps to 15 and 9 min, respectively. The final assay used 450 μL of sample input volume and consisted of the following steps: (1) centrifugation (-15 min), total nucleic acid extraction and DNase treatment for isolation of total RNA (-1 h), (2) cDNA synthesis with ribosomal RNA (rRNA) depletion (-1 h), (3) barcoded adapter ligation, library PCR amplification and purification on an automated instrument (-6.5 h), (4) library pooling (-5 min), (5) Illumina (San Diego, CA) sequencing (5 or 13 h, depending on whether a MiniSeq or NextSeq sequencer is used), and (6) bioinformatics analysis for viral detection and quantification using the SURPI+ pipeline (-1 h). Overall sample-to-answer assay turnaround time was 14–24 h. We used MS2 phage and External RNA Controls Consortium (ERCC) RNA Spike-In Mix (Invitrogen, Waltham, MA) added into each sample as internal qualitative and quantitative controls, respectively. The MS2 phage and ERCC sequencing results were also used to evaluate and interpret the background level in the sample, generally originating from the human host (Supplementary Tables 1 and 2). A commercial reference panel (Accuplex Panel, SeraCare, Milford, MA) consisting of quantified SARS-CoV-2, influenza A, influenza B, and respiratory syncytial virus (RSV) was spiked into pooled virus-negative

nasopharyngeal swab matrix as an external positive control (PC) for the assay (see Methods for details), with pooled virus-negative nasopharyngeal swabs from healthy uninfected donors as the negative matrix serving as an external negative control (NC).

The SURPI+ computational pipeline, run as a container on either a server or cloud, was used for the identification of viral respiratory pathogens from mNGS data[21,22]. Three enhancements were made (Fig. 2A). First, we added the capability for viral load quantification using the PC and a standard curve generated for each sample from the ERCC. A standard curve is generated for each sample using the normalized ERCC results and absolute quantification by comparison of the ERCC data with the external PC. Second, "tagging" of Genbank accession numbers in the SURPI+ database was incorporated to allow inclusion of curated viral reference genomes, such as those deposited in the FDA-ARGOS database[23], for virus identification by alignment and results reporting. Third, a custom algorithm consisting of de novo assembly of metagenomic reads and translated nucleotide or amino acid alignment of the reads to a viral protein database was developed to enable detection of novel, sequence-divergent viruses[23].

Following clinical chart review, we investigated the correlation between viral load concentration, quantified in copies per milliliter (cp/mL) (Fig. 2B), and infection severity, which was categorized on a scale ranging from asymptomatic to mild, moderate, and severe (Supplementary Table 3). We observed significant differences in median viral loads between patients with asymptomatic or mild and moderately severe to severe infections ($P < 0.001$ by the Mann-Whitney U test) (Fig. 2B, left). Further stratification of patients into asymptomatic, mild, moderate, and severe infections highlighted an increasing trend in viral load concentrations (Fig. 2B, right), with significant differences in median viral loads overall across all severity levels ($P < 0.001$ by Kruskal-Wallis H test). Pairwise differences in median viral loads between asymptomatic or mild and moderately severe infections infections were also significant ($P < 0.01$ by Mann-Whitney U test).

Quality control metrics were based on those previously established for a validated cerebrospinal fluid mNGS assay[21] and included a minimum of 5 million preprocessed reads per sample, >75% of data with quality score >30 (Q > 30), and successful detection of the internal spiked MS2 phage control and all four respiratory viruses in the PC. A threshold criterion of ≥3 non-overlapping viral reads or contigs aligning to the target viral genome was considered a positive detection. Overall, 93% (156 of 167) of both positive ($n = 111$) and negative ($n = 56$) nasopharyngeal swab samples met QC metrics; those that did not meet QC metrics were excluded from the analysis.

### Analytical sensitivity
We adopted Clinical and Laboratory Standards Institute guidelines for NGS-based infectious diseases testing (MM24)[24] and validation of multiplex nucleic acid assays (MM17)[25] to conduct a comprehensive evaluation of assay performance metrics (Table 1). To determine limits of detection (LoD), negative nasopharyngeal swab matrix was spiked with the Accuplex Verification Panel and diluted at concentrations ranging from 5000 to 100 copies/mL, with 10 to 40 replicates at each concentration. By 95% probit analysis, the LoD was determined for each of the four representative organisms in the panel (SARS-CoV-2, Influenza A, Influenza B, and RSV). We found LoDs ranging from 439 to 706 copies/mL for the four respiratory viruses in the positive control (Fig. 3). The achieved average LoD of 550 copies/mL was comparable within one log to reported LoDs from specific reverse transcription-polymerase chain reaction (RT-PCR) assays for detection of viral respiratory pathogens[26].

### Linearity
To evaluate the assay's capability to accurately quantitate viral load for detected viruses, a linearity panel was generated using five log

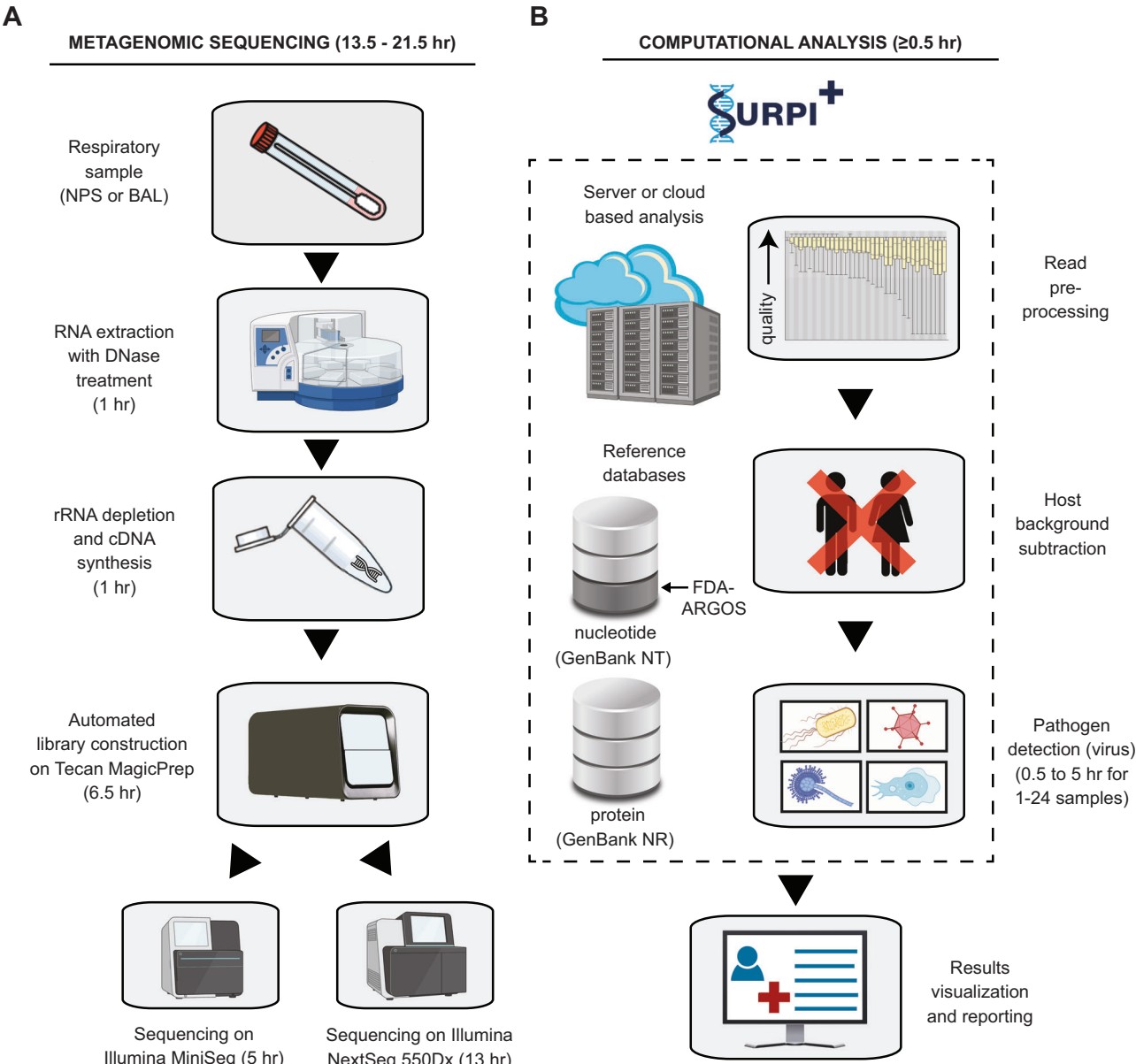

**A**  METAGENOMIC SEQUENCING (13.5 - 21.5 hr)

Respiratory sample (NPS or BAL)

RNA extraction with DNase treatment (1 hr)

rRNA depletion and cDNA synthesis (1 hr)

Automated library construction on Tecan MagicPrep (6.5 hr)

Sequencing on Illumina MiniSeq (5 hr)

Sequencing on Illumina NextSeq 550Dx (13 hr)

**B**  COMPUTATIONAL ANALYSIS (≥0.5 hr)

SURPI+

Server or cloud based analysis

Reference databases

FDA-ARGOS

nucleotide (GenBank NT)

protein (GenBank NR)

Read pre-processing

Host background subtraction

Pathogen detection (virus) (0.5 to 5 hr for 1-24 samples)

Results visualization and reporting

**Fig. 1 | Schematic of the mNGS assay workflow. A** RNA from respiratory samples is extracted and treated with DNase. Internal control is added to assess human background during sequencing. Human rRNA is depleted during cDNA synthesis. Libraries are generated on the automated Tecan MagicPrep NGS instrument. Libraries are normalized, pooled, and loaded onto the sequencer. **B** Sequences are processed using SURPI+ software for alignment and classification. Reads are pre-processed by trimming of adapters and removal of low-quality/low-complexity sequences, followed by computational subtraction of human reads. Reads are mapped to the closest matched genome to identify non-overlapping regions using NCBI GenBank and FDA-ARGOS database. To aid in analysis, automated result summaries, heat maps of both raw and normalized read counts, and coverage and pairwise identity plots are generated for clinical interpretation. Total turnaround time is between 14 and 22 h depending on type of sequencer used.

dilutions of a quantified high-titer SARS-CoV-2 positive nasal swab sample and compared to a commercially available AccuSpan™ HCV RNA Linearity Panel. For both panels, the calculated linearity was 100% after running duplicates or triplicate replicates across a minimum of four 10-fold dilutions (Fig. 4). The absolute $\log_{10}$ deviation of calculated from expected viral loads was <0.52 $\log_{10}$, which was favorable in comparison to the interquartile ranges for virus-specific qPCR assays between different laboratories[27].

**Precision**
We measured intra-assay precision by testing two PC and two NC samples within the same run using different barcodes across 20 runs and inter-assay precision by testing 20 PC and 20 NC samples using

different barcodes across 20 separate runs. Essential agreement (EA) was 100% and intra- and inter-assay precision were within our a priori established limits of <10% and <30% log-transformed coefficients of variation in reads per million, respectively (Table 1).

**Inclusivity and exclusivity**
To evaluate the ability of the mNGS assay to detect a wide range of targets (inclusivity), we obtained commercially available culture supernatants from 17 respiratory viruses representing different sublineages and subspecies. Viruses were spiked into negative control matrix at concentrations ranging from $1.3 \times 10^3$ to $1.2 \times 10^7$ 50% tissue culture infective dose (TCID50) per mL in a 1:10 ratio (Table 2). All 17 (100%) of 17 viruses in these contrived samples were correctly

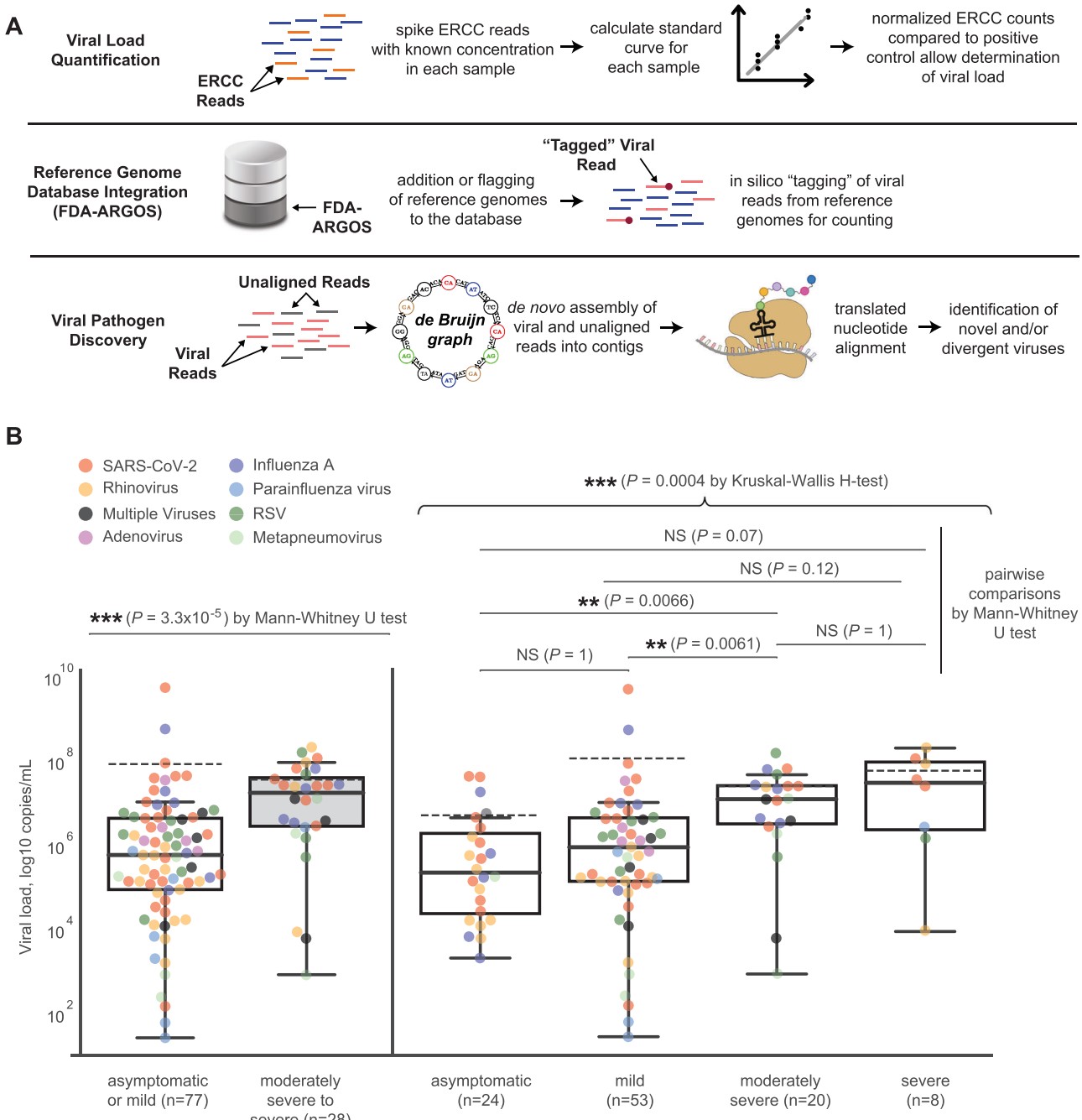

**Fig. 2 | Enhancements to the SURPI+ bioinformatics pipeline for pathogen identification. A** Schematic diagram of modifications made to the SURPI+ bioinformatics pipeline to enhance its pathogen detection capabilities. The modifications include (1) calculation of the estimated viral load for each detected virus in the sample using a quantitative internal spiked ERCC control (top row), (2) incorporation of reference-grade databases such as the FDA-ARGOS database by"tagging" of GenBank accession numbers in the SURPI+ database (middle row), and (3) identification of novel, sequence-divergent viruses using de novo viral genome assembly and translated nucleotide (amino acid) alignments to a viral protein database (bottom row). **B** Pairwise and overall comparisons of viral load medians among groups stratified by severity: asymptomatic ($n = 24$), mild ($n = 53$), moderately ($n = 20$), or severe ($n = 8$) respiratory infection. For the box and whiskers plots, the solid line within each box represents the median log viral load, while the dashed line indicates the mean log viral load. The interquartile range (IQR) is shown by the height of the box, with whiskers extending to the minimum and maximum values within 1.5 times the IQR. Each point corresponds to a detected virus, with different colors representing different virus species or genera. Mann-Whitney U and Kruskal-Wallis H tests were used for pairwise and overall significance testing, respectively. All tests were two-sided with Bonferroni correction for multiple comparisons, and the significance level was set at 0.05.

identified by mNGS assay at the sublineage or subspecies level. Additionally, we identified subtypes of rhinovirus and enterovirus from PCR-positive clinical samples that were not differentiated by multiplex RT-PCR (Fig. 5A). We also evaluated the ability of the mNGS assay to identify uncommon or rare viral pathogens associated with respiratory infections ($n = 8$ virus-positive tracheal aspirate samples) or central nervous system (CNS) infections ($n = 4$ cerebrospinal fluid samples) in severely ill hospitalized patients (Table 2 and Fig. 5B). The assay detected 11 (100%) of 11 viruses in these samples. To assess the exclusivity of the mNGS assay, we spiked two mixtures of

**Table 1 | Performance characteristics of the UCSF viral respiratory mNGS assay**

| Metrics | Method | Expected Target | | Results | |
|---|---|---|---|---|---|
| Limit of detection (LoD) | Detection of PC dilution by probit analysis | <1000 copies/mL | | Target<br>SARS-CoV-2<br>Influenza A<br>Influenza B<br>RSV | LoD<br>439 copies/mL<br>706 copies/mL<br>493 copies/mL<br>563 copies/mL |
| Linearity | Correlation of PC with assay quantification | $R^2 > 90\%$ | | $R^2 = 100\%$ | |
| Precision | Intra-Assay: PC and NC within the same run across 20 runs. | Concordance 100% EA | Log-transformed CV <10% | Concordance 100% EA | Log-transformed CV <10% |
| | Inter-Assay: PC and NC across 20 separate runs | 100% EA | <30% | 100% EA | <30% |
| Inclusivity | Detection of viruses from diluted culture supernatant | 100% detection | | 100% detection (17/17) | |
| | Detection of viruses in positive BAL/CSF diluted samples | 100% detection | | 100% detection (11/11) | |
| Exclusivity | Detection of viruses in known organism mixtures[a] | No false-positive | | No false-positive | |
| Contamination | Detection of cross-contamination on the sample wells | No carryover contamination | | Cross-contamination of 0.1% between adjacent wells but no carryover contamination | |
| Interference | Detection of PC spiked with hemolytic blood | Detection at all concentrations | | Detection at all concentrations | |
| | Detection of PC spiked with Human RNA | Detection at all concentrations | | Detection at all concentrations | |
| | Detection of PC spiked with bacterial DNA/RNA | Detection at concentration ≤ $10^7$ cells/mL | | Detection at concentration ≤ $10^7$ cells/mL | |
| | Detection of virus-positive overtly mucoid BAL samples | Detection in all BAL samples | | Target detected in 13/14 (92.9%) valid sample runs | |
| Stability | Detection of targets in samples held at 4 °C for 7 days or after 3 freeze-thaw cycles | 100% concordance | | 100% concordance | |
| Accuracy | Detection in virus positive and negative samples (n = 191) | Sensitivity > 90%<br>Specificity > 90%<br>Accuracy > 90%<br>PPA > 90%<br>NPA > 90% | | Original testing<br>Sensitivity: 93.6%<br>Specificity: 93.8 %<br>Accuracy: 93.7 % | After discrepancy testing and clinical adjudication<br>PPA: 98.7%<br>NPA: 98.1%<br>Overall: 97.9% |
| Detection of divergent viruses | Detection by an in silico analysis of divergent viruses (n = 70) | Sensitivity > 95%<br>Specificity > 95% | | Sensitivity: 98.6%<br>Specificity: 100% | |

*PC* positive control consisting of 4 respiratory viruses spiked into pooled nasopharyngeal swab matrix, *IC* spiked internal control consisting of a RNA MS2 phage, *NC* negative control, *EA* essential agreement, *CV* coefficient of variation, *PPA* positive percent agreement, *NPA* negative percent agreement.
[a]Two mixtures were assessed. The first mixture included detectable concentrations of CMV, HIV, *Klebisella pneumoniae*, *Streptococcus agalactiae*, *Aspergillus niger*, *Cryptococcus neoformans* and *Toxoplasma gondii*, and corresponds to positive control material from a previously validated CSF assay[21]. The second mixture was a commercial reference panel, the ZymoBIOMICS Microbial Community Standard (Zymo Research, Tustin, CA), and consisted of 10 bacterial and fungal pathogens at varying concentrations (*Listeria monocytogenes*—12%, *Pseudomonas aeruginosa*—12%, *Bacillus subtilis*—12%, *Escherichia coli*—12%, *Salmonella enterica*—12%, *Lactobacillus fermentum*—12%, *Enterococcus faecalis*—12%, *Staphylococcus aureus*—12%, *Saccharomyces cerevisiae*—2%, and *Cryptococcus neoformans*—2%) that were spiked into negative nasopharyngeal swab matrix.

microorganisms, including a previously reported positive control mNGS panel consisting of 7 representative pathogens[21] and a commercial reference panel consisting of 10 bacterial and fungal species, into negative nasopharyngeal swab matrix and analyzed multiple aliquots (Table 1 and Supplementary Table 4). Detected reads from non-viral pathogenic organisms did not result in any false-positive detections for viral pathogens.

**Contamination, matrix effect, and stability**
We evaluated potential cross-contamination between nearby sample wells and carryover contamination across successive runs from 10 SARS-CoV-2 high-titer clinical samples and 24 controls (cycle threshold, or $C_t = 16$–20) loaded in a modified checkerboard pattern (with at least one space between samples) on a 96-well plate, to mimic a single run on the Illumina NextSeq instrument. Only one possible cross-contamination event was observed, with a single SARS-CoV-2 read detected in one of the negative control wells at a subthreshold reporting level. We also evaluated the effects of interference from potential interfering substances, human RNA, and bacterial DNA/RNA on mNGS assay performance. Hemolysis, lipids, bilirubin, and human genomic RNA spiked into PC matrix at concentrations of 0.1–100 μg/mL did not interfere with respiratory virus detection, but bacterial

DNA/RNA spiked into PC matrix at concentrations ≥$1 \times 10^7$ cells/mL resulted in failure to detect viruses due to high background. To evaluate the potential matrix effect from samples with high host background, we analyzed 14 PCR-positive highly mucoid bronchoalveolar lavage (BAL) samples obtained from lung transplant or cystic fibrosis patients undergoing surveillance bronchoscopy (Supplementary Table 5). All 14 samples had high host background, and 13 (92.9%) of 14 samples had very high host background. As a result, 6 (42.9%) of 14 samples had neither detection of the internal spiked MS2 phage control nor of a respiratory virus, and thus excluded from further analysis, as they not pass equencing quality control criteria (Supplementary Table 1). The respiratory viral pathogen was detected in all (100%) of the remaining 8 samples. We concluded that highly mucoid samples can inhibit the assay due to high host background. Finally, we evaluated mNGS assay stability; qualitative detection was not affected by keeping samples for up to 7 days at 4 °C or subjecting the samples to 3 freeze/thaw cycles.

**Accuracy**
To evaluate accuracy, 191 residual samples after routine clinical testing were obtained from the UCSF Clinical Microbiology Laboratory, including 110 virus-positive samples (104 upper respiratory swab

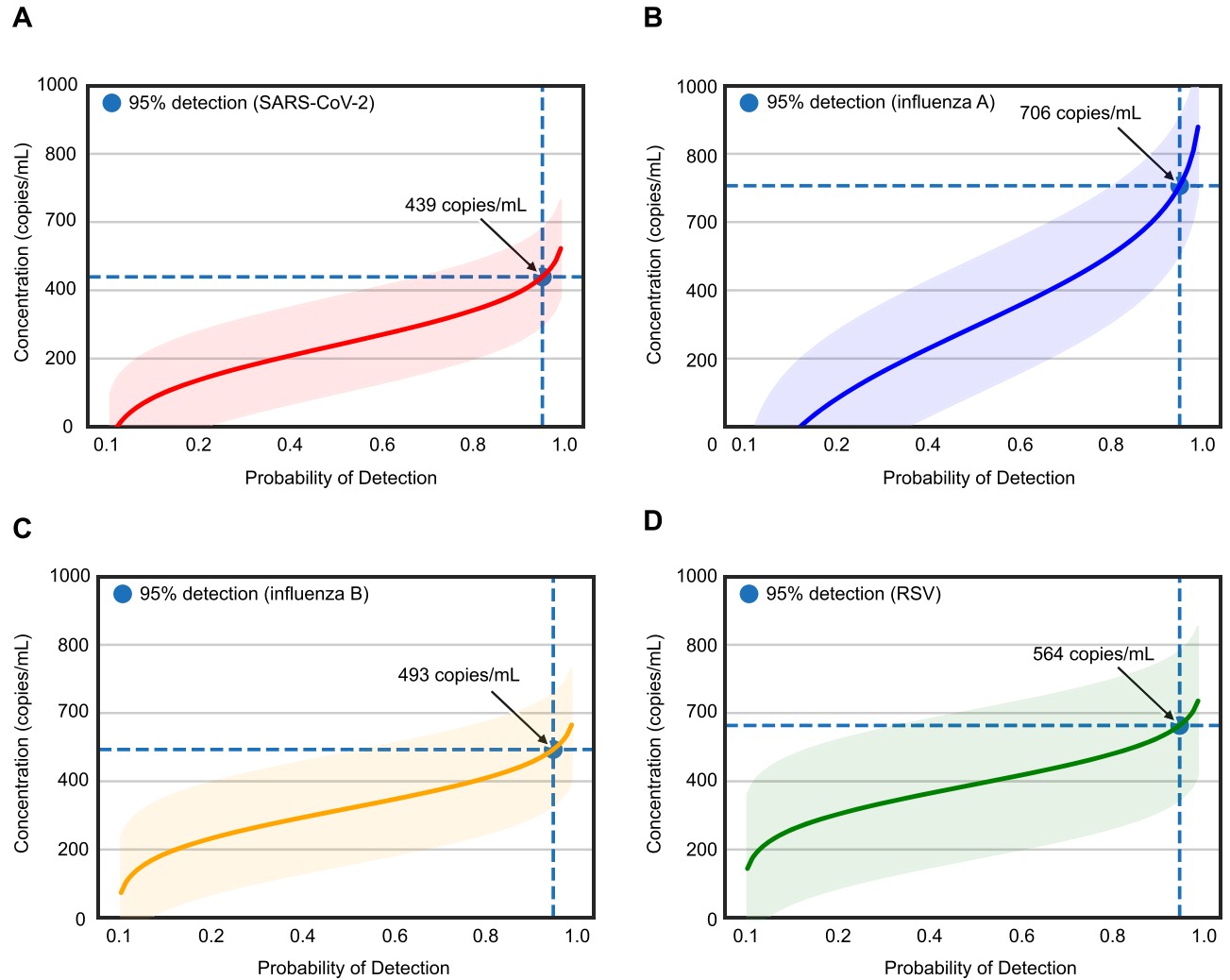

**Fig. 3 | Limits of detection (LoD) study.** Probit regression analysis curves plotting the viral titer in copies/mL (y-axis) against the calculated detection probability (x-axis) of (**A**) SARS-CoV-2, (**B**) influenza A, (**C**) influenza B and (**D**) respiratory syncytial virus (RSV). The regression curves and error bands (surrounding shaded areas) representing 95% confidence intervals for each curve as determined by probit regression analysis are shaded in a different color for each virus. The probability of detection corresponding to 95% is denoted with a blue circle for each virus. Probit analyses were done using Python software (version 3.7.12). Results show a LoD ranging from 439 to 706 copies/mL for the 4 respiratory viruses in the positive control.

samples and 6 BAL fluids) from patients with acute respiratory infection (Supplementary Data 1), along with 81 virus-negative samples (52 upper respiratory swab samples and 29 BAL fluids) (Fig. 6). As more than one target may be positive with mNGS and respiratory viral multiplex panel (RVP) testing using FDA-approved in vitro diagnostic assays, sensitivity/specificity analyses were performed by assessing each result independently to assign true/false-positive/negative calls (see Methods for details). Compared to results from RVP RT-PCR testing, the mNGS assay exhibited 93.6% (103 of 110) sensitivity, 93.8% (76 of 81) specificity, and 93.7% (179 of 191) accuracy.

Discrepancy testing and clinical adjudication (DTCA) of 14 mNGS positive-RVP negative samples using blinded chart review by two board-certified infectious diseases physician (PB and CYC) and orthogonal assays run by the California Department of Public Health Viral and Rickettsial Disease Laboratory confirmed the presence of 9 respiratory viruses missed by RVP, allowing them to be reclassified as true positives (Supplementary Table 6). Viruses detected by mNGS but not targeted by RVP were not considered false-positive results. In one case, while the original RVP and orthogonal PCR testing returned negative results, mNGS identified rhinovirus C with high confidence. A

review of the viral sequences revealed 12 non-overlapping reads across the human rhinovirus C genome (Fig. 7A, B). Cross-contamination was ruled out, as no other sample in the sequencing batch tested positive for rhinovirus. A nucleotide BLAST (blastn) search confirmed sequences with high homology (95–98% identity) to known rhinovirus C strains. Although the exact primer binding sites for the clinical RT-PCR assays used in the current study are unknown, we identified, for the rhinovirus C sample, the presence of mismatches in primer and probe regions from previously reported RT-PCR assays targeting the 5′-untranslated region (UTR)[28,29] (Fig. 7C), which explained the detection by mNGS despite negative RT-PCR results.

Similarly, DTCA was performed on the 7 mNGS negative/RVP positive samples along with repeating the RVP assay (if possible, on a different instrument). This reassessment resulted in 5.5 samples being reclassified as true negatives (1 sample harbored two organisms adjudicated as one true negative and one false negative) (Supplementary Table 7). Compared to a composite standard that incorporates discrepancy testing and clinical adjudication, positive, negative, and overall predictive agreements of the mNGS assay were 98.7% (110.5 of 113), 98.1% (76.5 of 78), and 97.9% (187 of 191), respectively.

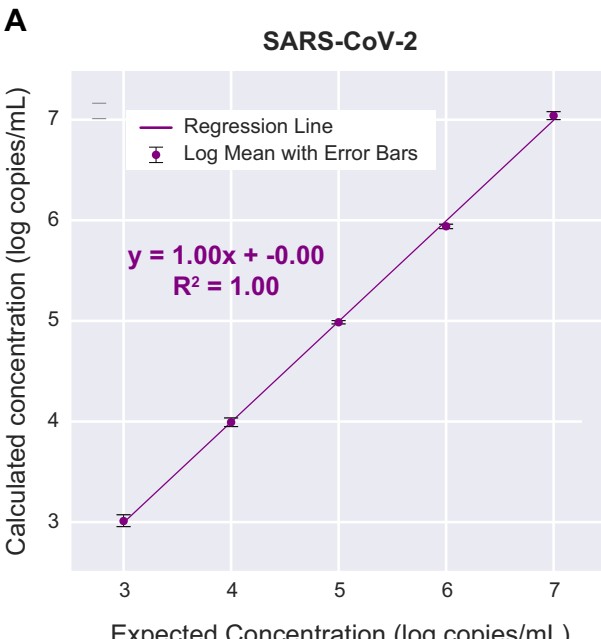

**A** SARS-CoV-2

$y = 1.00x + -0.00$
$R^2 = 1.00$

**B** Hepatitis C virus

$y = 0.91x + 0.04$
$R^2 = 1.00$

**Fig. 4 | Evaluation of linearity and viral load quantification for the mNGS assay. A** A quantified SARS-CoV-2 PCR-positive nasopharyngeal swab from a patient with COVID-19 was serially diluted in donor nasal swab matrix and tested across 4 $\log_{10}$ dilutions. **B** A quantified HCV PCR-positive plasma sample from a patient with hepatitis C infection was serially diluted in donor plasma and tested across 4 $\log_{10}$ dilutions. At each dilution, the calculated mean concentration from three replicates is plotted against the expected concentration on a log scale, and the $R^2$ correlation coefficient is determined by linear regression.

## Detection of novel, sequence-divergent viruses

To benchmark the capability of the modified SURPI+ pipeline for detection of novel, highly divergent viruses in silico, we created a simulated sequencing output file containing many known human viral pathogens of clinical and public health significance, including those with pandemic potential (Fig. 8A). We then removed all viral reference sequences of the same type (for example, all human polyomaviruses, coronaviruses, or parainfluenza viruses) or corresponding to the same genus or species from the SURPI+ 2019 reference database. Next, we used the SURPI+ pipeline to analyze the simulated sequencing file against both the original and "filtered" reference databases. In this analysis, 98.6% (69 of 70) of human viruses were detected at a sequencing depth of 100 reads per million (RPM) and 100% (70 of 70) at 1000 RPM based on homology to known animal or plant viruses (Fig. 8B). Of note, bunyaviruses pathogenic to humans, which are among the most divergent viruses, were still identified by translated nucleotide (amino acid) alignment to plant viruses (for example, detection of Venezuelan equine encephalitis virus based on homology to vanilla latent virus).

## Discussion

We validated a clinical mNGS assay in a CLIA laboratory as a Laboratory Developed Test (LDT) for agnostic viral respiratory pathogen detection intended to aid in patient diagnosis and public health surveillance. Our main goal was to develop, optimize, and streamline a protocol for respiratory viral mNGS testing that could be deployed and run routinely in clinical or public health laboratories. The mNGS assay developed here has favorable performance characteristics compared to clinical RVP testing, including a limit of detection of ~500 copies/mL, viral load quantification with 100% linearity, and sensitivity, specificity, and accuracy ranging from 93.6–93.8%. However, in contrast to targeted assays such as RVP, the mNGS assay is capable of detecting, in principle, all known as well as novel viral pathogens in respiratory samples. In addition, mNGS assay performance was found to be superior to RVP (97.9% versus 95.0% overall agreement) after discrepancy testing and clinical adjudication. The correlations we observed between viral load and disease severity highlight the potential for complementary quantitative viral load measurements to aid in distinguishing beween asymptomatic infection or colonization and overt severe respiratory disease, thereby informing clinical management and treatment, as has been previously demonstrated for certain non-respiratory viruses such as CMV[30]. Following completion

### Table 2 | Detection of a broad range of viruses in contrived samples

| Contrived Sample Type | Correctly Identified Virus by mNGS Assay | |
|---|---|---|
| Positive cerebrospinal fluid (CSF) spiked in negative matrix | Lymphocytic choriomeningitis virus (LCMV) | |
| | Herpes simplex virus 2 (HSV-2) | |
| | Varicella-zoster virus (VZV) | |
| | Herpes simplex virus 1 (HSV-1) and Epstein-Barr virus (EBV) | |
| Positive bronchoalveolar lavage (BAL) spiked in negative matrix | Parainfluenza virus Type 4 | Parechovirus A |
| | Influenza C virus | Human bocavirus |
| | Primate bocaparvovirus 1 | Coronavirus 229E |
| | Coronavirus NL63 | |
| Viral culture fluid spiked in negative control matrix (1:10) | Adenovirus Type 1 | Coronavirus 229E |
| | Coronavirus NL63 | Coxsackie virus Type A1 |
| | Echovirus | Human metapneumovirus 16 |
| | Influenza B virus | Measles virus |
| | Mumps virus | Parainfluenza virus type 2 |
| | Parainfluenza virus type 3 | Parainfluenza virus type 4A |
| | Parechovirus type 1 | Rhinovirus A16 |
| | Rhinovirus B14 | Rubella virus |
| | Influenza B virus | |

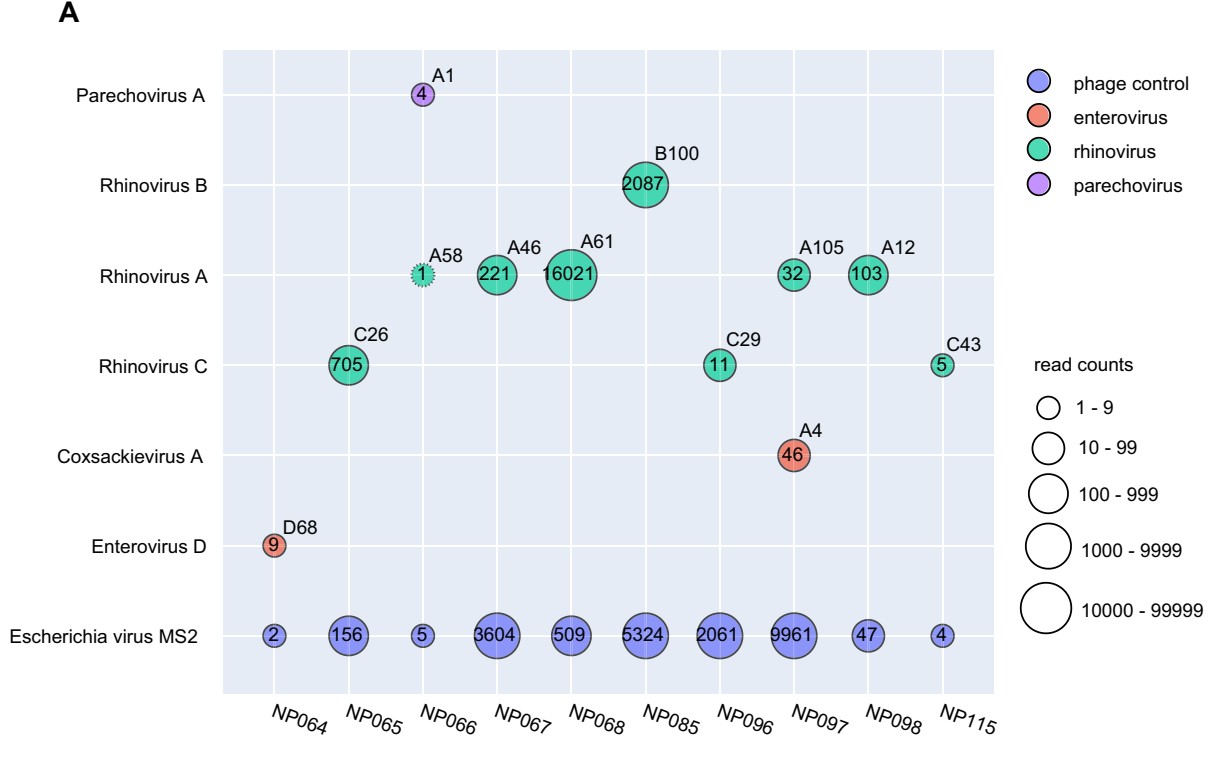

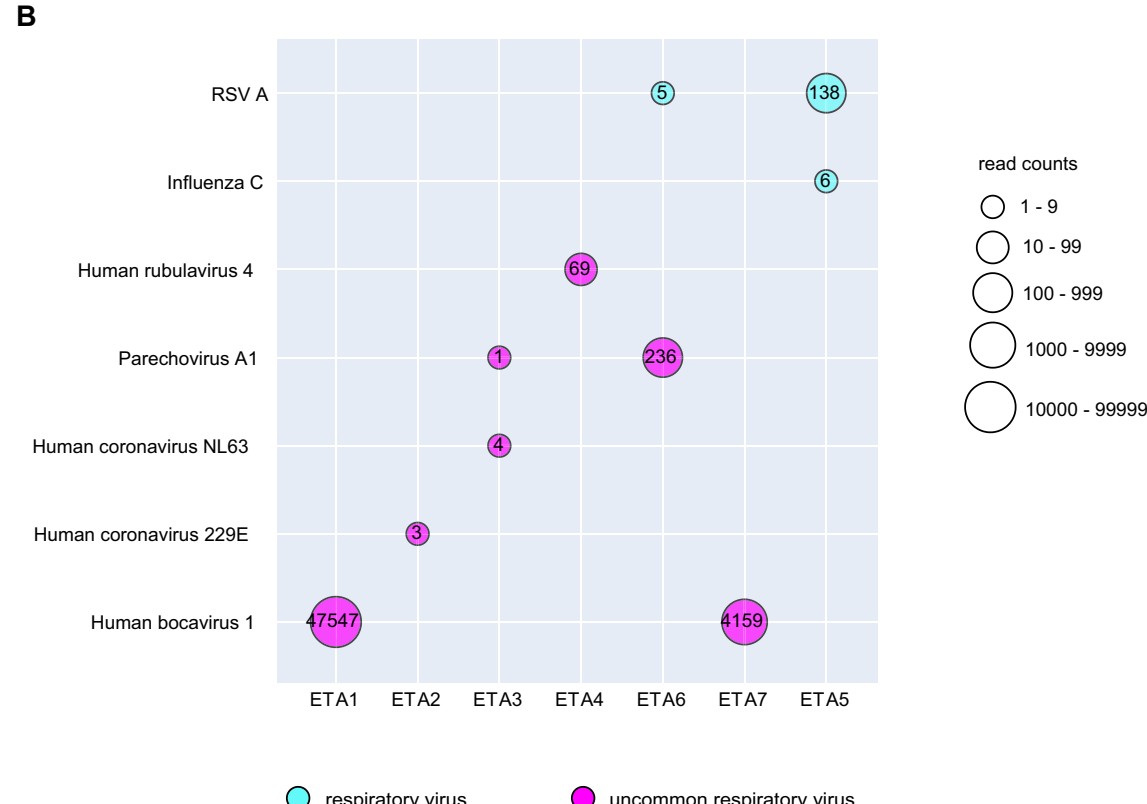

**Fig. 5 | Demonstration of inclusivity and clinical use cases for the mNGS assay. A** Genotyping of rhinovirus and enterovirus subtypes from PCR-positive nasal swab samples. Conventional clinical multiplex RT-PCR tests do not distinguish between rhinoviruses and enteroviruses, nor are they able to subtype more pathogenic strains such as rhinovirus C or enterovirus D68 in association with acute flaccid myelitis[34,35]. **B** Detection of uncommon or rare viral pathogens causing respiratory infections in critically ill mechanically ventilated hospitalized patients. The circles correspond to detected viruses and are color-coded by virus and scaled by read counts. For each detected virus, the read count is shown in the circle, while the identified genotype after SURPI+ pipeline is shown in the upper right quadrant. Abbreviations: ETA endotracheal aspirate.

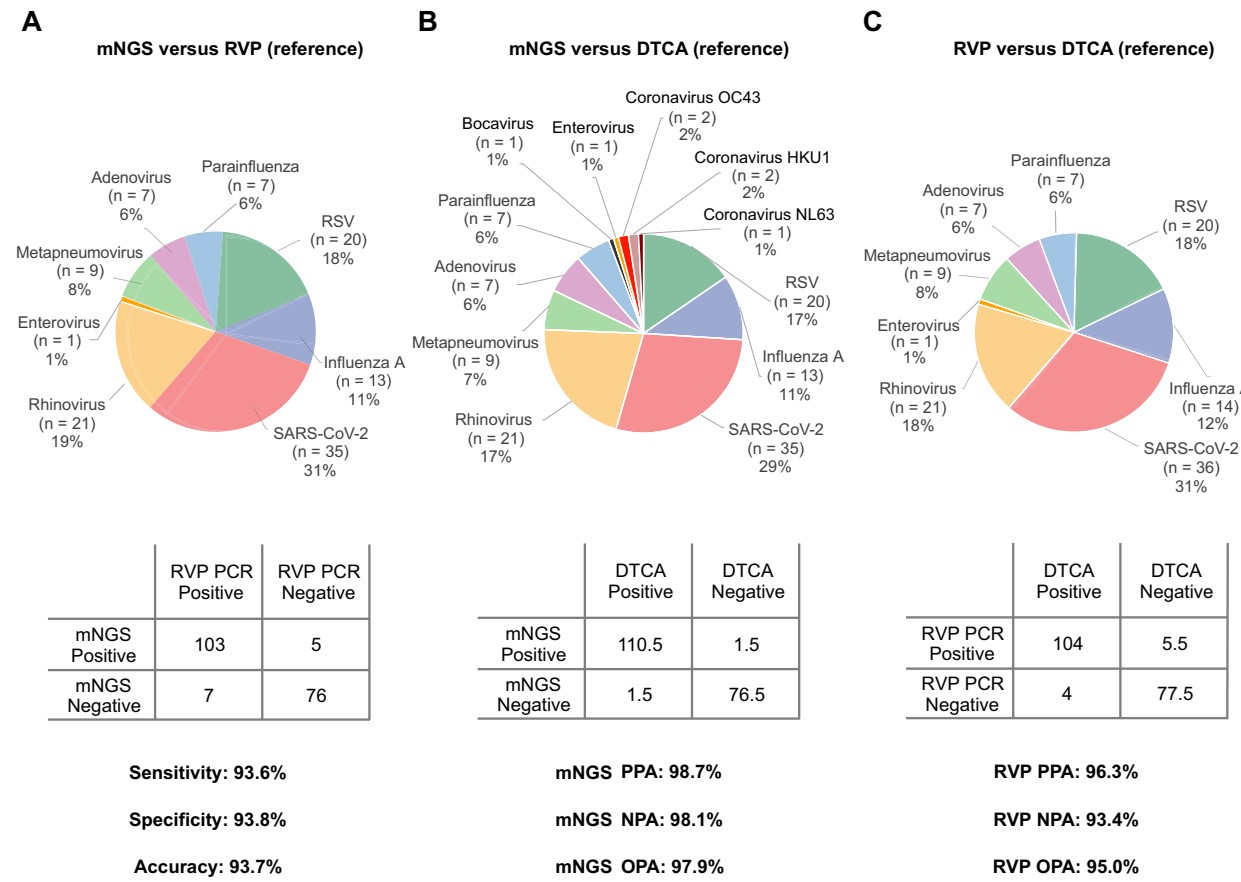

**Fig. 6 | Accuracy evaluation for the mNGS assay.** Pie charts and 2 × 2 contingency tables showing the distribution of detected viruses and performance metrics. **A** mNGS against RVP testing, (**B**) mNGS testing against DTCA, and (**C**) RVP testing against DTCA. RVP testing using FDA IVD assays includes detection of respiratory syncytial virus, parainfluenza viruses 1–3, metapneumovirus, rhinovirus/enterovirus, influenza A virus, influenza B virus, and adenovirus. Discrepant samples that were mNGS-positive/RVP-negative or mNGS-negative/RVP-positive underwent orthogonal testing by targeted virus-specific PCR at the state public health laboratory and medical chart review for the most likely diagnosis by clinical adjudication. Abbreviations: mNGS metagenomic next-generation Sequencing, PCR polymerase chain reaction, RVP viral respiratory panel, DTCA discrepancy testing and clinical adjudication, PPA positive percent agreement, NPA negative percent agreement, OPA overall percent agreement, RSV respiratory syncytial virus, FDA Food and Drug Administration, IVD in vitro diagnostic.

of the validation, our assay received breakthrough device designation from the US Food and Drug Administration (FDA) in August of 2023. Widespread implementation of highly accurate, rapid mNGS assays such as this, with enhanced capacity to detect novel viruses, will support robust preparation for and rapid responses to the next viral pandemic.

Speed is a critical factor for diagnosis of respiratory infections, especially in critically ill patients with lower respiratory involvement and in outbreak investigations of novel or emerging viruses with pandemic potential. Here we also aimed to develop an assay that could be deployable widely in clinical and public health laboratories. Thus, we optimized many of the steps of the mNGS assay and moved the key RNA/cDNA library preparation step to an automated platform, the MagicPrep NGS system (Tecan Genomics, Inc., Männedorf, Switzerland). We further demonstrated that sequencing can be performed on the Illumina MiniSeq using the Rapid Reagent Kit for a faster 5-h turnaround time or on the Illumina NextSeq 550Dx using the Mid-Output Reagent Kit for a 13-h turnaround time, depending on laboratory needs and priorities. All together, these modifications resulted in an assay with a turnaround time of 14–24 h and <2 h of hands-on technician time.

Orthogonal testing and clinical adjudication performed on discordant results demonstrated that the RVP assay is an imperfect gold standard with which to judge mNGS performance. The mNGS assay was able to not only detect uncommon infections from viruses not covered on existing RVP panels, but also, in multiple cases, detect viruses that are detectable by RVP in principle but tested negative. Unlike RVP, mNGS does not rely on specific primers or probes and is hence less susceptible to primer failure due to viral evolution, as evidenced by the mNGS positive and RVP negative rhinovirus case presented here. Thus, RVP assay sensitivity will likely decrease over time by continual viral mutations, which is an inevitable feature of SARS-CoV-2 and many other RNA viruses[31]. Notably, a previous study evaluating the usefulness of published PCR primers in detecting rhinovirus infection reported that none of the published rhinovirus-specific PCR primer pairs could detect all human rhinoviruses in 101 genotyped clinical specimens[32]. In addition, broader sampling of the viral genome by mNGS may result in increased sensitivity of virus detection compared to RVP due to increased robustness to variability in the relative levels of viral gene expression in infected cells[33]. Most of the false-negative mNGS samples were confirmed as true negative after chart review and repeating the RVP assay. Most likely, these represented false-negative results during the original RVP run, either due to low viral titers associated with high cycle thresholds (>36) or degradation of samples over ime and/or after multiple freeze-thaw cycles.

In the study, we used several approaches to demonstrate the capacity of the mNGS assay to identify novel and/or emerging viruses with divergent genomes. The assay was successful in detecting uncommon and unusual viral pathogens associated with both severe respiratory infections from bronchoalveolar lavage fluid and central

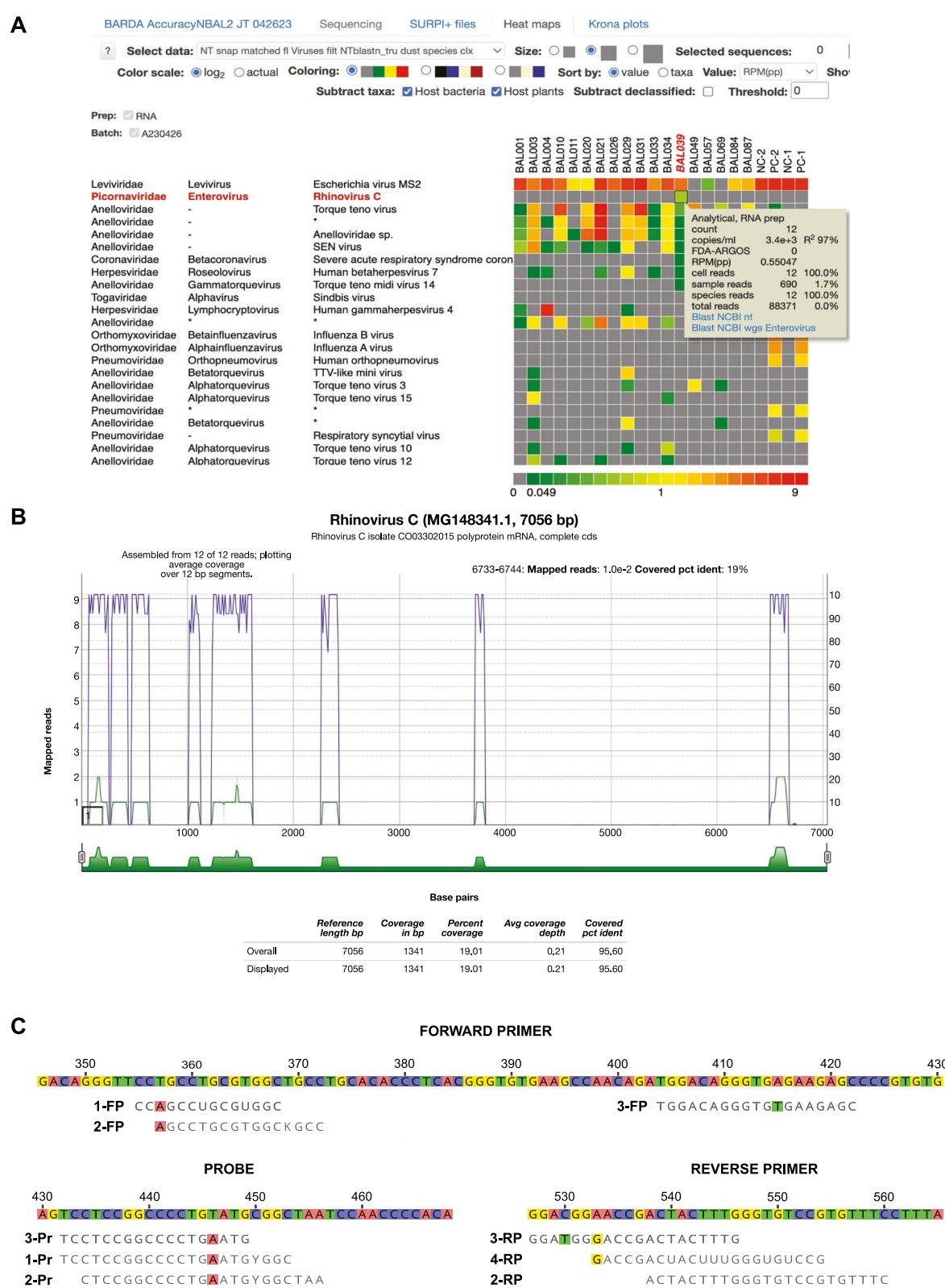

nervous infections from contrived CSF samples. mNGS testing also enabled subtyping of specific viral strains with increased virulence, such as enterovirus D68, which has been linked to acute flaccid myelitis in children[34,35], and rhinovirus C, which has been associated with invasive pulmonary and bloodstream infection in immunocompromised patients[36,37]. Importantly, the mNGS assay was also able to detect DNA viruses, such as adenovirus and bocavirus, in both clinical and

contrived samples, despite the incorporation of DNase treatment in the protocol. Detection of DNA viruses is presumably based on detection of transcribed viral mRNA in infected cells, although may also enabled by incomplete DNA digestion from the DNase enzyme.

To evaluate the capacity for mNGS testing to identify novel viruses using a modified SURPI+ computational pipeline, we performed an in silico analysis of a contrived metagenomic dataset consisting of

**Fig. 7 | In-depth analysis of a rhinovirus C detection by mNGS that was discrepant with RT-PCR. A** A heat map generated from SURPI+ analysis shows 12 reads aligning to rhinovirus C from a single sample, excluding the possibility of cross-contamination. Each column denotes a clinical sample, while each row corresponds to a taxonomic identification at the species, genus, or family level. The asterisks refer to "declassification" of reads from one level to the next higher taxonomic level (for example, from species to genus). **B** A coverage map shows that the 12 reads span the genome of the most closely matched rhinovirus C genome in the reference database identified by SURPI+ (accession number MG148341.1)

without overlap, with coverage of 19% of the ~7000 base pair (bp) genome. **C** Several mismatches in the primer and probe sequences from published RT-PCR assays targeting the 5′-untranslated region (5′-UTR) are observed when compared to the viral mNGS reads, providing a likely explanation for the discrepant mNGS and RT-PCR results. The four assays are labeled 1 through 4 and correspond to Lu et al.[29] (1), Tapparel et al.[47] (2), Gunson et al.[48] (3), and Steininger et al.[49] (4). The mismatched nucleotides are highlighted with a background colour. Note that the assay from Steininger, et al. does not include a probe. Abbreviations: FP forward primer, Pr probe, RP reverse primer.

reads from the genomes of human viruses of pandemic potential spiked into background using a reference database depleted of all known human viral sequences. This analysis was done to simulate whether "novel" human viruses with pandemic potential could be identified based on homology to known plant and animal viruses. All 70 of the human viral pathogens tested were successfully identified, including those with distant homology to other viruses. Indeed, chikungunya virus, in the *Alphavirus* genus of the *Togaviridae* family, was only identified after removal of all alphavirus sequences because of distant homology to vanilla latent virus in the family *Alphaflexiviridae*. Notably, alphaflexiviruses contain a distinct lineage of alphavirus-like replication proteins that lack a recognized protease domain[38]. These in silico results demonstrate that the pipeline is able to detect highly diverse viruses from families that are known to be potentially pathogenic to humans and that emerge from animal reservoirs (for example, *Bunyaviridae, Flaviviridae*, and *Adenoviridae*). If a novel, highly divergent virus from an uncharacterized family were detected, with little to no homology to any viral reference sequence, much more work would be needed to ascertain its clinical significance, or whether it is even capable of infecting humans, including formal assessment of Koch's postulates with modificatons by Rivers for causality[39].

Our validation study has limitations. First, we tested very few bronchoalveolar lavage fluid samples from patients with acute respiratory infection ($n = 6$) and very few clinical samples harboring rare or unusual respiratory viruses ($n = 7$). Further validation of assay performance with these kinds of samples is needed. Second, mNGS testing was performed exclusively on samples from US patients, so viral pathogen diversity may not be representative of all populations globally. Third, we did not formally prove that the mNGS assay would be able to detect a novel, sequence-divergent virus, but instead demonstrated the ability of the test to detect such a virus using an in silico analysis, an approach which nonetheless has been used in previous studies to benchmark mNGS bioinformatic pipelines for viral pathogen discovery[40,41]. Finally, we did not address the utility of the mNGS assay for routine diagnosis in patients with unexplained infections or for outbreak surveillance in public health. Both efforts will likely require future prospective clinical and/or epidemiologic investigation.

In our study, the raw materials and labor costs for running the mNGS validation samples were ~$300 USD per sample (Supplementary Table 8). However, this represents a lower limit for costs and does not account for costs related to assay implementation, bioinformatics analysis and director review, proficiency testing, quality and regulatory management, incomplete batch testing, the use of different sequencers (for example, NextSeq versus MiniSeq), and sample accessioning/reporting, among others. Thus, the actual costs for running the assay in clinical and/or commercial laboratories are much higher. In contrast, the estimated costs for running RVP assays in our clinical laboratory range from $100–$150 USD per sample. Nevertheless, the benefits for mNGS testing of greatly expanded scope of detection, capability to identify novel emerging viruses, and comparable performance likely outweigh the costs under certain clinical and public health scenarios. Further investigations that include cost-benefit analyses are needed to identify clinical use cases and indications for viral respiratory mNGS testing.

Even though the mNGS assay described here has exhibited high performance characteristics for sensitivity and specificity for the detection of viral pathogens, it is currently unlikely to replace multiplex RVP assays as a first-line test, as these panels are inexpensive and have more rapid turnaround times than mNGS. In addition, RVP assays easy to perform, with self-contained instrumentation that does not require batching and some platforms being CLIA-waived for use in point of care settings. However, mNGS testing could be particularly useful in public health laboratories that are more likely to receive and test samples from patients infected with unusual or novel viruses that are not part of the standard RVP testing panels. Of note, a modified protocol based on the assay was used to identify adeno-associated virus 2 in co-infections with adenoviruses and herpesviruses in cases of acute severe hepatitis in children as part of a nationwide US outbreak[42]. The respiratory mNGS assay developed here could also be implemented as a second-line test in clinical laboratories for patients with presumed viral bronchiolitis and pneumonia when RVP testing is negative. This strategy would be useful for diagnosis of rare and/or unexpected infections in immunocompromised patients or returning travelers, for whom there is a wider differential diagnosis.

## Methods
### Human sample collection
Residual laboratory-confirmed virus-positive upper respiratory swab or BAL samples from clinical patient testing were retrieved from the UCSF Clinical Microbiology Laboratory. Acceptable upper respiratory swab samples included (1) bilateral nasopharyngeal swabs, (2) bilateral anterior nares swabs, (3) oropharyngeal swabs, (4) combined nasopharyngeal and oropharyngeal swabs, and (5) combined oropharyngeal/mid-turbinate nasal swabs. All samples were required to meet minimal sample handling, storage, and volume requirements for inclusion in our study. Samples were stored at 4 °C for <24 h before being de-identified, aliquoted, and stored in -80 °C freezer prior to mNGS processing, thus undergoing one freeze-thaw cycle.

### Inclusion and ethics
All samples meeting minimal volume (≥450 μL), sample handling (at most one freeze-thaw step), and storage (kept frozen at −80 °C) requirements were included in this study. Samples along with clinical and laboratory metadata were collected according to a biobanking protocol with waiver of consent approved by the UCSF Institutional Review Board (protocol no. 11-05519)

### External controls preparation
The external positive control (PC) was prepared by spiking a pooled negative nasal swab matrix with a commercially available reference material, the Accuplex Verification Panel (SeraCare, Milford, MA). This panel consisted of a mixture of non-infectious SARS-CoV-2, influenza A, influenza B, and RSV genomes encapsidated in a synthetic protein coat to mimic the structure of a viral capsid. This PC material was "spiked in" at a titer of ~$10^4$ copies/mL for each virus control, 1–2 logs higher than the estimated limit of detection of the assay (~500 copies/mL). The negative matrix was prepared by pooling nasopharyngeal swab samples from asymptomatic individuals and was used as an external negative control (NC).

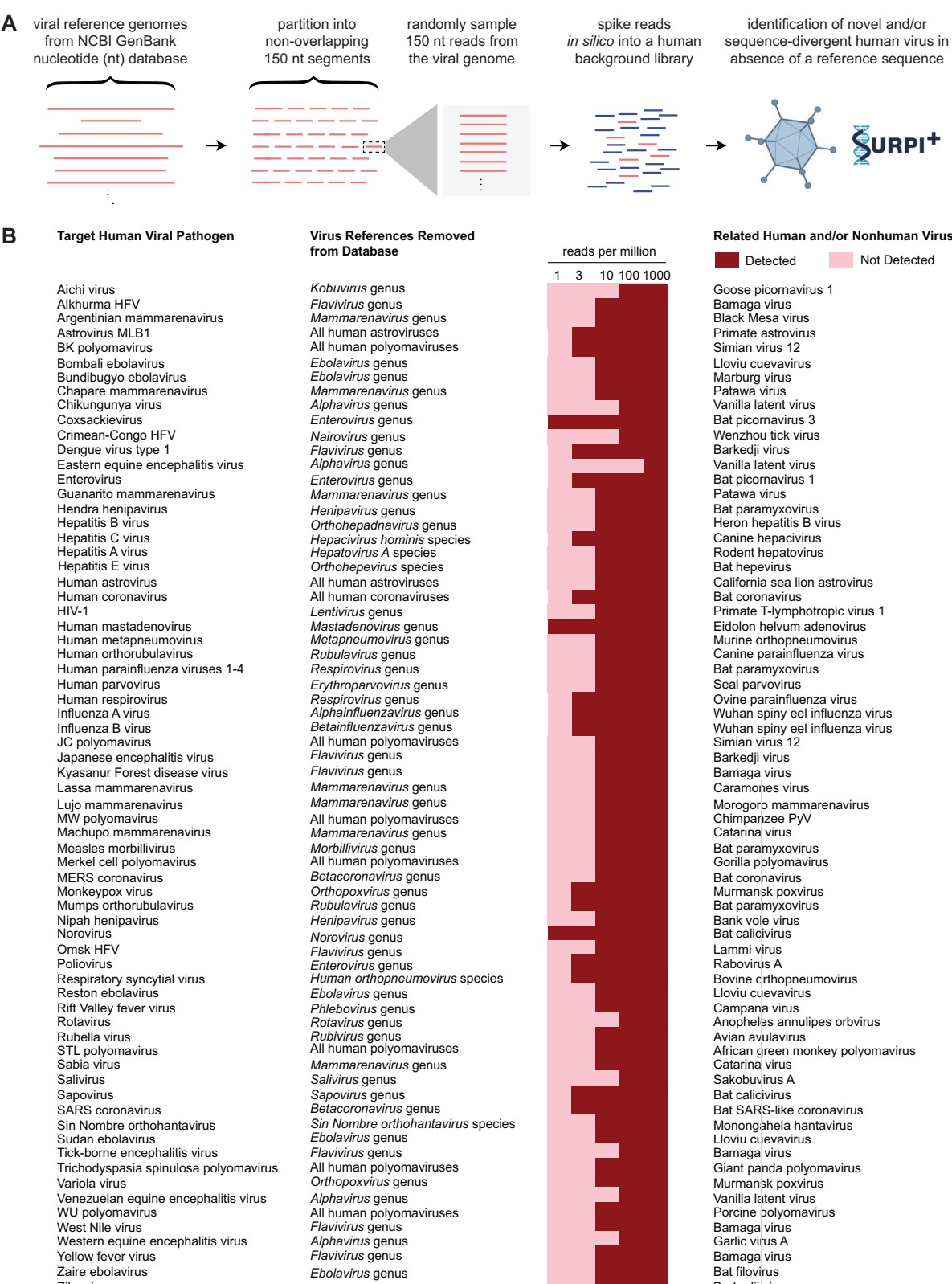

## Nucleic acid extraction

500 µL of upper respiratory swab or BAL fluid was centrifuged at 16,000 × g for 10 min. The MagMAX™ Viral/Pathogen II (MVP II) Nucleic Acid Isolation Kit (cat # A48383, Thermo Fisher Scientific, Waltham, MA) and the KingFisher™ Flex Purification System with a 96 deep-well head (Thermo Fisher Scientific, Waltham, MA) were used for total nucleic acid extraction. This protocol was modified to include DNase treatment using TURBO™ DNase (cat # AM2238, Thermo Fisher Scientific, Waltham, MA) as a host depletion step during extraction. Bacteriophage MS2 (cat # 22-156-880, Zeptometrix, Buffalo, NY) was added to all samples including the negative control as an internal qualitative control.

**Fig. 8 | In silico demonstration of novel, sequence-divergent virus detection using the mNGS assay. A** Representative viral reference genomes corresponding to outbreak viruses of clinical and public health significance with pandemic potential are retrieved from the NCBI GenBank database, partitioned into non-overlapping segments, and then randomly sampled and spiked in silico into a negative nasal swab matrix sequencing library. A higher-level set of taxonomic identifiers (species, genus, and/or family) corresponding to these viruses is removed from the SURPI+ reference dataset and the simulated sequencing file is analyzed using both the original and "restricted reference" databases. **B** Viruses can be detected using the modified SURPI+ pipeline despite lacking a taxonomic reference at levels down to 10–100 reads per million (RPM). Abbreviations: EEEV Eastern equine encephalitis virus, ERCC External RNA Controls Consortium, FDA-ARGOS FDA dAtabase for Reference Grade micrObial Sequences, HFV hemorrhagic fever virus, HIV human immunodeficiency virus, JCPyV JC polyomavirus, PC positive control, PyV polyomavirus, TSPyV trichodysplasia spinulosa polyomavirus, SURPI+ sequence-based ultrarapid pathogen identification, VEEV Venezuelan equine encephalitis virus, WEEV Western equine encephalitis virus.

## Library preparation and sequencing

Simultaneous reverse transcription of purified RNA, spiked in with ERCC RNA controls (cat # 4456740, Invitrogen, Waltham, MA), and ribosomal RNA (rRNA) depletion were carried out using NEBNext® Ultra™ II RNA First Strand Synthesis Module (cat #s E7771S/ E7771L, New England Biolabs, Ipswich, MA) and QIAseq FastSelect-rRNA HMR Kit (cat # 334385, Qiagen, Germantown, MD), respectively, followed by second strand cDNA synthesis using Sequenase™ Version 2.0 DNA Polymerase (cat # 70775Z1000UN, Thermo Fisher Scientific, Waltham, MA). Complementary DNA (cDNA) was purified using AMPure XP beads (cat # A63881, Beckman Coulter, Brea, CA) and loaded on the MagicPrep NGS instrument (Tecan Genomics, Inc., Männedorf, Switzerland) to undergo end-repair, adapter ligation, and barcoding, amplification (25 cycles) and purification using the DNA-Seq Mech kit (cat #s 30186627/30186628/30186629, Tecan Genomics, Inc., Männedorf, Switzerland). Libraries were quantified and normalized using the Qubit dsDNA HS Assay (cat # Q32854, Thermo Fisher Scientific, Waltham, MA) on the Qubit Flex (Thermo Fisher Scientific, Waltham, MA). Final pooled libraries were sequenced as single-end reads on either the Illumina (San Diego, CA) MiniSeq using the Rapid Reagent Kit (100 cycles) or on the Illumina NextSeq 550 using the Mid-Output or High-Output Kit (150 cycles).

## Bioinformatics

The SURPI+ computational pipeline, run as a container (v1.0.0) on either a secure server or cloud infrastructure, was used for identification of respiratory viral pathogens from mNGS data. Reads were pre-processed by trimming of adapters and removal of low-complexity and low-quality sequences, followed by computational subtraction of human reads. The Scalable Nucleotide Alignment Program[43] nucleotide aligner was run using an edit distance of 16 against the National Center for Biotechnology Information (NCBI) nucleotide (NT) database (March 2019, with inclusion of the SARS-CoV-2 WuHan-Hu-1 genome accession number NC_045512), which was pre-filtered to retain only viral reads. The pipeline was modified to include "tagging", or annotation, of entries from reference sequences that constitute a subset of the NCBI NT database, such as FDA-ARGOS[23]. Note that the FDA-ARGOS database, while quality controlled and regulated, contains only 1428 microbial strains, the majority of which are bacterial. It had also not been updated with recent viruses such as SARS-CoV-2; thus, this study did not detect any reads matching to viral genomes in FDA-ARGOS. The pipeline was modified to accommodate additional reference databases as needed such as GISAID[44]. The pipeline was also modified to use SPAdes (v3.15.4)[45] and DIAMOND (v2.0.15)[46], respectively, for optional de novo assembly of reads into contiguous sequences (contigs) and translated nucleotide sequence alignment for identification of sequence-divergent viruses. Viral reads were identified using DIAMOND at a e-value cutoff of $10^{-5}$. Coverage maps were automatically generated by mapping SURPI+ -classified viral reads to the most likely reference genome.

Quality control metrics for the assay were based on those previously established for cerebrospinal fluid[21], and include a minimum of 5 million preprocessed reads per sample, >75% of data with quality score >30 ($Q > 30$), and successful detection of the 4 respiratory viruses in the PC and the internal spiked MS2 phage control. A criterion of ≥3 non-overlapping viral reads or contigs aligning to the target viral genome was considered a positive detection.

## Evaluation of mNGS analytical performance characteristics

The automated standard operating procedures and sequencing runs for these clinical validation studies were performed by a California state-licensed clinical laboratory scientist. LoD was determined for each of the four representative organisms in the PC by probit analysis using a series of dilutions ranging from 100 to 5,000 copies/mL, with 10 to 40 replicates at each concentration. Linearity was demonstrated by plotting the standard curve. To validate the quantification using the ERCC and the positive control, we serially diluted an HCV positive plasma to known concentration ranging from $4 \times 10^6$ to $4 \times 10^3$ copies/mL in triplicate. We then compared the quantitative measure to the known measure. Precision was determined using repeat analysis of two PC and two NC samples across 20 runs (intra-assay reproducibility) and by testing 20 PC and 20 NC samples across 20 separate runs (inter-assay reproducibility). To assess inclusivity, commercially available cultured supernatants were obtained to assess the assay's ability to detect the intended targets. Each of the 17 respiratory viruses, with titers ranging from $1.3 \times 10^4$ to $1.2 \times 10^8$ TCID50/mL, were spiked into the negative control matrix at a 1:10 dilution. These viruses represented known sublineages and subspecies and we evaluated the ability of the assay to detect the virus. We also tested samples of confirmed virus-positive BAL ($n = 7$) and CSF samples ($n = 4$) spiked into negative matrix to evaluate assay performance with respect to detection of unusual viruses. To assess the exclusivity of the mNGS assay, we spiked a previously established mixture of seven representative pathogenic organisms to determine the false positive detection rate for viral pathogens. We evaluated cross-contamination between adjacent sample wells and carryover contamination across successive runs from samples with high viral loads. Interference was determined using PC spiked with known amounts of hemolytic blood, lipids, bilirubin, human RNA, and bacterial DNA/RNA. The effect of mucus in BAL positive fluids was also assessed. Stability was determined by keeping samples for up to 7 days at 4 °C or subjecting the samples to 3 freeze/thaw cycles. Accuracy was determined using 191 clinical samples comprising 110 virus-positive samples (103 upper respiratory swab samples and 7 BAL fluids) from patients with acute respiratory infection, along with 81 virus-negative samples (52 upper respiratory swab samples and 29 BAL fluids). Samples were obtained from patients at the University of California, San Francisco (UCSF). The viral RT-PCR comparator assays that were used include the Genmark ePlex (Carlsbad, CA), Luminex NxTAG (Austin, TX), and/or Luminex Verigene RP Flex Respiratory Pathogen Panels. mNGS results were compared with original clinical testing and then with a composite reference standard including discrepancy testing and clinical adjudication. In the second comparison, when results were discordant, orthogonal testing was performed using a different instrument or an independent CLIA laboratory (the California Department of Public Health) in addition to clinical adjudication to reclassify mNGS results. The second comparison was reported as positive percent agreement (PPA) and negative percent agreement (NPA), as selective discrepancy testing can bias sensitivity and specificity results.

**Orthogonal discrepancy testing at the California Department of Public Health**

Specimens were tested by real-time PCR based on CDC protocols using a viral respiratory panel, an unpublished CDPH laboratory-developed test (LDT). Viruses that can be detected by this panel included human metapneumovirus, respiratory syncytial virus, adenovirus, parainfluenza virus (types 1, 2, 3, and 4), enterovirus/rhinovirus, and human coronaviruses 229E, OC43, NL63, and HKU1.

**In silico analysis for identification of novel, sequence-divergent viruses using the SURPI+ pipeline**

To assess detection capability for novel, sequence-divergent viruses, an in silico analysis was performed. Representative viral reference genomes corresponding to outbreak viruses of clinical and public health significance with pandemic potential were retrieved from the NCBI GenBank database, partitioned into non-overlapping segments, and then randomly sampled and spiked in silico into a negative nasal swab matrix sequencing library. We then took a higher-level set of taxonomic identifiers (species, genus, and/or family) corresponding to these viruses and removed all entries with these taxonomic identifiers from the SURPI+ reference dataset. Next, we used the SURPI+ pipeline to analyze the simulated sequencing file against both the original and "restricted reference" databases and evaluated the performance of the pipeline in detecting "simulated" novel and/or divergent viruses that lacked a reference sequence.

**Statistical analyses**

Statistical analyses were performed using scipy (version 1.5.3) and rstatix (version 0.7.0) packages as implemented in Python (version 3.7.12) and R (version 4.0.3), respectively. The non-parametric Mann-Whitney U test was used for pairwise comparisons of viral load medians, while the Kruskal-Wallis H test was used for comparisons of medians across all severity groups. Probit regression analyses were done using scipy (version 1.5.3), numpy (version 1.19.1), and statsmodels (version 0.12.2) as implemented in Python software (version 3.7.12).

Sensitivity and specificity analyses were performed as follows: as more than one target may be positive with mNGS and RVP, each result was independently assessed in every sample and true/false-negative/positive were accordingly assigned to each result. However, the total number of observations was kept constant (one sample = one observation = 1). For instance, in the case a test detected two organisms, namely the culprit pathogen and a contaminant, the former was assigned 0.5 true-positive and the latter 0.5 false-positive, such that their sum was always equal to 1. In addition, as we used RVP as a comparator which included only a limited number of targets, mNGS positive-RVP negative results that were not a target for the RVP were not considered as false-positive results.

**Reporting summary**

Further information on research design is available in the Nature Portfolio Reporting Summary linked to this article.

## Data availability

Human-subtracted raw sequence data were submitted to the Sequence Read Archive (SRA) database. (BioProject accession number PRJNA1084017 and umbrella BioProject accession number PRJNA171119). Source data are provided as a Source Data file. Sequence metadata is available in a Zenodo data repository (https://zenodo.org/doi/10.5281/zenodo.10553378). Source data are provided with this paper.

## Code availability

Custom scripts and code for data analyses and visualization are available in a Zenodo data repository (https://zenodo.org/doi/10.5281/zenodo.10553378). The SURPI+ bioinformatics pipeline is described in prior publications[21,22]. The code for SURPI+ includes proprietary algorithms for taxonomic classification, filtering, and pathogen software that have been filed under US patent 11380421, "Pathogen detection using next generation sequencing". Pleae contact the University of California Office of Technology Management regarding access to and use of the software.

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

## Acknowledgements

We thank the staff at the UCSF Clinical Microbiology Laboratory for help in collecting nasopharyngeal swab and bronchoalveolar lavage fluid samples. This work was financially supported in part by BARDA EZ-BAA award 75A50122C00022 (C.Y.C.), US CDC grants 75D30122C15360 and 75D30121C12641 (C.Y.C.), Abbott Laboratories (C.Y.C.), and the Chan-Zuckerberg Biohub (C.Y.C.). The funders had no role in the design and conduct of the study; collection, management, analysis, and interpretation of the data; preparation, review or approval of the manuscript; and decision to submit the manuscript for publication. The content of this paper is solely the responsibility of the authors and does not represent the official views or opinions of the National Institutes of Health, Kaiser Permanente, California Department of Public Health or the California Health and Human Services Agency. Use of trade names and commercial sources is for identification only and does not imply endorsement by the California Department of Public Health or the California Health and Human Services Agency. Figures 1A, B, 2A, and 8A were created in part using images from BioRender.com released under a Creative Commons Attribution-NonCommercial-NoDerivs 4.0 International (CC-BY-NC-ND) license.

## Author contributions

C.Y.C conceived of and designed the study. J.K.T., V.S., D.S., J.S., N.S., A.F., H.J.H., J.N., M.O., N.B., J.T., D.I., B.F., H.R., M.H., C.M., D.A.W. and C.Y.C coordinated the sequencing efforts and laboratory studies. J.K.T., A.C., H.G. and S.Y. processed samples. J.K.T., V.S., D.S., E.K., A.C., H.G., S.Y., M.D.L., P.B. and C.Y.C. analyzed data. J.K.T., J.S., N.S., A.F., J.N., M.O., P.M.M. and C.R.L. collected samples. J.K.T., V.S., E.K., P.B., M.D.L and C.Y.C. wrote the manuscript. J.K.T., V.S., E.K., P.B., and C.Y.C. prepared the figures. J.K.T., V.S., D.S., E.K., N.S., A.F., H.J.H., J.N., M.O., N.B., J.T, D.I., B.F., H.R., M.H., D.A.W., P.M.M., C.R.L., M.D.L., P.B. and C.Y.C edited the manuscript. J.K.T., V.S., E.K., M.D.L., P.B. and C.Y.C. revised the manuscript. All authors read the manuscript and agree to its contents.

## Competing interests

C.Y.C. is a founder of Delve Bio and on the scientific advisory board for Delve Bio, Flightpath Biosciences, Biomeme, Mammoth Biosciences, BiomeSense and Poppy Health. He is also an inventor on US patent 11380421, "Pathogen detection using next generation sequencing", under which algorithms for taxonomic classification, filtering, and pathogen detection are used by SURPI+ software. C.Y.C. receives research support from Delve Bio and Abbott Laboratories, Inc. The other authors declare no competing interests.

## Additional information

Jessica Karielle Tan [1,2,8], Venice Servellita[1,2,8], Doug Stryke[1,2,8], Emily Kelly[1], Jessica Streithorst[1], Nanami Sumimoto[1,2], Abiodun Foresythe[1,2], Hee Jae Huh[1,2,3], Jenny Nguyen[1,2], Miriam Oseguera [1,2], Noah Brazer[1,2], Jack Tang[1,2], Danielle Ingebrigtsen[1], Becky Fung[1], Helen Reyes[1], Melissa Hillberg[1], Alice Chen[4], Hugo Guevara[4], Shigeo Yagi[4], Christina Morales[4], Debra A. Wadford [4], Peter M. Mourani [5], Charles R. Langelier [6,7], Mikael de Lorenzi-Tognon [1,2], Patrick Benoit[1,2] & Charles Y. Chiu [1,2,6,7] ✉

[1]Department of Laboratory Medicine, University of California San Francisco, San Francisco, CA, USA. [2]Abbott Pandemic Defense Coalition, Abbott Park, IL, USA. [3]Department of Laboratory Medicine and Genetics, Samsung Medical Center, Sungkyunkwan University School of Medicine, Seoul, South Korea. [4]Viral and Rickettsial Disease Laboratory, Center for Laboratory Sciences, California Department of Public Health, Richmond, CA, USA. [5]Department of Pediatrics, University of Arkansas for Medical Sciences, Little Rock, AR, USA. [6]Division of Infectious Diseases, Department of Medicine, University of California San Francisco, San Francisco, CA, USA. [7]Chan-Zuckerberg Biohub, San Francisco, CA, USA. [8]These authors contributed equally: Jessica Karielle Tan, Venice Servellita, Doug Stryke. ✉e-mail: charles.chiu@ucsf.edu

