## [Peer Review File · Nature Communications]

Laboratory validation of a clinical metagenomic next-generation sequencing assay for respiratory virus detection and discoveryEditorial Note: This manuscript has been previously reviewed at another journal. This document only contains reviewer comments and rebuttal letters for versions considered at *Nature Communications*.

REVIEWERS' COMMENTS

Reviewer #1 (Remarks to the Author):

1. In the discussion, consider adding that current multiplex respiratory panels are also easy to perform, self-contained instrumentation, do not require batching with some version being CLIA-waived for use at the point-of-care.
2. Cost analysis. Consider providing a range for the mNGS similar to the range provided for the RVP since the cost accounting provided is focused on the use of the NextSeq. Would testing on the MiniSeq be more expensive? Also, what is the cost of SURPI? This is particularly important given what commercial labs currently charge for mNGS assay and the cost of \$300 might be a bit misleading.
3. Thank you for providing the data showing such significant correlation between viral loads and disease severity. One issue that often comes with respiratory samples compared to more homogeneous samples like blood/plasma, is the level of variability in the fluid collection. Is it possible to calibrate the viral load to an internal control using mNGS to help normalize? I would recommend adding some comments in the discussion to that effect.

Reviewer #2 (Remarks to the Author):

Authors have nicely addressed all of my comments. This is a very nice study with clinical utility.

RESPONSE TO REVIEWERS

Reviewer #1 (Remarks to the Author):

1. In the discussion, consider adding that current multiplex respiratory panels are also easy to perform, self-contained instrumentation, do not require batching with some version being CLIA-waived for use at the point-of-care.

We added this statement as requested by the reviewer –

In addition, RVP assays easy to perform, with self-contained instrumentation that does not require batching and some platforms being CLIA-waived for use in point of care settings.

2. Cost analysis. Consider providing a range for the mNGS similar to the range provided for the RVP since the cost accounting provided is focused on the use of the NextSeq. Would testing on the MiniSeq be more expensive? Also, what is the cost of SURPI? This is particularly important given what commercial labs currently charge for mNGS assay and the cost of \$300 might be a bit misleading.

We are very hesitant to provide a range for mNGS costs as suggested by the reviewer because the costs overall can be highly variable and are dependent on a number of factors. We would prefer to present just a single cost analysis for the costs of running the assay for purposes of our validation. The actual clinical and commercial testing costs will likely be significantly different. To minimize the risk that the costs cited will be misleading we added a statement that these are the raw materials and labor costs for running the assay as part of the validation and do not reflect what the clinical costs of running the assay would be in the revised manuscript (see below):

In our study, the raw materials and labor costs for running the mNGS validation samples were ~\$300 USD per sample (Supplementary Table 8). However, this represents a lower limit for costs and does not account for costs related to assay implementation, bioinformatics analysis and director review, proficiency testing, quality and regulatory management, incomplete batch testing, the use of different sequencers (for example, NextSeq versus MiniSeq), and sample accessioning / reporting, among others. Thus, the actual costs for running the assay in clinical and/or commercial laboratories are much higher. In contrast, the estimated costs for running RVP assays in our clinical laboratory range from \$100-\$150 USD per sample. Nevertheless, the benefits for mNGS testing of greatly expanded scope of detection, capability to identify novel emerging viruses, and comparable

performance likely outweigh the costs under certain clinical and public health scenarios. Further investigations that include cost-benefit analyses are needed to identify clinical use cases and indications for viral respiratory mNGS testing.

3. Thank you for providing the data showing such significant correlation between viral loads and disease severity. One issue that often comes with respiratory samples compared to more homogeneous samples like blood/plasma, is the level of variability in the fluid collection. Is it possible to calibrate the viral load to an internal control using mNGS to help normalize? I would recommend adding some comments in the discussion to that effect.

The current assay already “calibrates the viral load to an internal control using mNGS” to help normalize. The internal control that is used are the spiked ERCC (External RNA Controls Consortium reads). The normalization is described in the Methods and legend to Figure 2:

From Methods:

First, we added the capability for viral load quantification using the PC and a standard curve generated for each sample from the ERCC. A standard curve is generated for each sample using the normalized ERCC results and absolute quantification by comparison of the ERCC data with the external PC.

From Figure 2 legend:

Figure 2 . Enhancements to the SURPI+ bioinformatics pipeline for pathogen identification.

(A) Schematic diagram of modifications made to the SURPI+ bioinformatics pipeline to enhance its pathogen detection capabilities. The modifications include calculation of the estimated viral load for each detected virus in the sample using a quantitative internal spiked ERCC control (top row)...

Reviewer #2 (Remarks to the Author):

Authors have nicely addressed all of my comments. This is a very nice study with clinical utility.